# Strong isolation by distance and evidence of population microstructure reflect ongoing *Plasmodium falciparum* transmission in Zanzibar

Sean V Connelly[1]*, Nicholas F Brazeau[1], Mwinyi Msellem[2], Billy E Ngasala[3,4], Ozkan Aydemir[5], Varun Goel[6], Karamoko Niaré[7], David J Giesbrecht[7], Zachary R Popkin-Hall[8], Chris Hennelly[8], Zackary Park[9], Ann M Moormann[5], John M Ong'echa[10], Robert Verity[11], Safia Mohammed[12], Shija J Shija[12], Lwidiko E Mhamilawa[3,4], Ulrika Morris[13], Andreas Mårtensson[4], Jessica T Lin[9], Anders Björkman[13,14], Jonathan J Juliano[9,15,16]†, Jeffrey A Bailey[7]†

[1]MD-PhD Program, University of North Carolina at Chapel Hill, Chapel Hill, United States; [2]Research Division, Ministry of Health, Zanzibar, United Republic of Tanzania; [3]Department of Parasitology and Medical Entomology, Muhimbili University of Health and Allied Sciences, Dar es Salaam, United Republic of Tanzania; [4]Global Health and Migration Unit, Department of Women's and Children's Health, Uppsala University, Uppsala, Sweden; [5]Department of Medicine, University of Massachusetts Chan Medical School, Worcester, United States; [6]Carolina Population Center, University of North Carolina at Chapel Hill, Chapel Hill, United States; [7]Department of Pathology and Laboratory Medicine, Brown University, Providence, United States; [8]Institute for Global Health and Infectious Diseases, School of Medicine, University of North Carolina at Chapel Hill, Chapel Hill, United States; [9]Division of Infectious Diseases, Department of Medicine, School of Medicine, University of North Carolina at Chapel Hill, Chapel Hill, United States; [10]Center for Global Health Research, Kenya Medical Research Institute, Kisumu, Kenya; [11]MRC Centre for Global Infectious Disease Analysis, Imperial College London, London, United Kingdom; [12]Zanzibar Malaria Elimination Program (ZAMEP), Zanzibar, United Republic of Tanzania; [13]Department of Microbiology, Tumor and Cell Biology, Karolinska Institutet, Stockholm, Sweden; [14]Department of Global Public Health, Karolinska Institute, Stockholm, Sweden; [15]Department of Epidemiology, Gillings School of Global Public Health, University of North Carolina at Chapel Hill, Chapel Hill, United States; [16]Curriculum in Genetics and Molecular Biology, University of North Carolina at Chapel Hill, Chapel Hill, United States

**\*For correspondence:**
sean_connelly@med.unc.edu

†co-senior authors

**Competing interest:** The authors declare that no competing interests exist.

## Abstract

**Background:** The Zanzibar archipelago of Tanzania has become a low-transmission area for *Plasmodium falciparum*. Despite being considered an area of pre-elimination for years, achieving elimination has been difficult, likely due to a combination of imported infections from mainland Tanzania and continued local transmission.

**Methods:** To shed light on these sources of transmission, we applied highly multiplexed genotyping utilizing molecular inversion probes to characterize the genetic relatedness of 282 *P. falciparum* isolates collected across Zanzibar and in Bagamoyo district on the coastal mainland from 2016 to 2018.

**Results:** Overall, parasite populations on the coastal mainland and Zanzibar archipelago remain highly related. However, parasite isolates from Zanzibar exhibit population microstructure due to the rapid decay of parasite relatedness over very short distances. This, along with highly related pairs within *shehias*, suggests ongoing low-level local transmission. We also identified highly related parasites across *shehias* that reflect human mobility on the main island of Unguja and identified a cluster of highly related parasites, suggestive of an outbreak, in the Micheweni district on Pemba island. Parasites in asymptomatic infections demonstrated higher complexity of infection than those in symptomatic infections, but have similar core genomes.

**Conclusions:** Our data support importation as a main source of genetic diversity and contribution to the parasite population in Zanzibar, but they also show local outbreak clusters where targeted interventions are essential to block local transmission. These results highlight the need for preventive measures against imported malaria and enhanced control measures in areas that remain receptive to malaria reemergence due to susceptible hosts and competent vectors.

**Funding:** This research was funded by the National Institutes of Health, grants R01AI121558, R01AI137395, R01AI155730, F30AI143172, and K24AI134990. Funding was also contributed from the Swedish Research Council, Erling-Persson Family Foundation, and the Yang Fund. RV acknowledges funding from the MRC Centre for Global Infectious Disease Analysis (reference MR/R015600/1), jointly funded by the UK Medical Research Council (MRC) and the UK Foreign, Commonwealth & Development Office (FCDO), under the MRC/FCDO Concordat agreement and is also part of the EDCTP2 program supported by the European Union. RV also acknowledges funding by Community Jameel.

## eLife assessment

Connelly and colleagues provide **convincing** genetic evidence that importation from mainland Tanzania is a major source of *Plasmodium falciparum* lineages currently circulating in Zanzibar. This study also reveals ongoing local malaria transmission and occasional near-clonal outbreaks in Zanzibar. Overall, the article effectively highlights the role of human movements in maintaining residual malaria transmission in an area targeted for intensive control interventions over the past decades and provides clear and **valuable** information for epidemiologists and public health professionals.

## Introduction

Malaria cases in Tanzania comprise 3% of globally reported cases, but transmission is heterogeneous, with the coastal mainland witnessing declining but substantial transmission of *Plasmodium falciparum* (*Alegana et al., 2021*; *World Health Organization, 2022*). On the other hand, the archipelago of Zanzibar is a pre-elimination setting, with low-level seasonal transmission (*Björkman et al., 2019*). This is largely due to the routine implementation of a combination of effective control measures, including robust vector control and routine access to effective antimalarials (*Björkman et al., 2019*). Despite these efforts, malaria has been difficult to eliminate from the archipelago. There are several reasons this may be the case: (1) frequent importation of malaria from moderate- or high-transmission regions of mainland Tanzania and Kenya (*Björkman et al., 2019*; *Le Menach et al., 2011*; *Lipner et al., 2011*; *Monroe et al., 2019*; *Morgan et al., 2020*; *Tatem et al., 2009*); (2) ongoing local transmission due to residual vector capacity despite strong vector control (*Björkman et al., 2019*); and (3) a reservoir of asymptomatic infections (*Björkman and Morris, 2020*; *Björkman et al., 2019*).

Parasite genomics has the potential to help us better understand malaria epidemiology by uncovering population structure and gene flow, providing insight into the changes in the parasite population including how parasites move between regions (*Neafsey et al., 2021*). Genomics has previously been used to study importation and transmission chains in other low-transmission settings in Africa and elsewhere (*Chang et al., 2019*; *Fola et al., 2023*; *Morgan et al., 2020*; *Patel et al., 2014*; *Roh et al., 2019*; *Sane et al., 2019*). Previously, we had investigated the importation of malaria into Zanzibar from the mainland using whole-genome sequencing, showing highly similar populations within the mainland and within the archipelago, but also identifying highly related parasite pairs between locations, suggesting a role for importation (*Morgan et al., 2020*). However, this work lacked sufficient

**Table 1.** Blood samples from Zanzibar and coastal Tanzania.

| Description | Location (district) | Dates | Clinical status* | Sample size | Age range (yr) | # in genome-wide analysis | # in drug resistance analysis |
|---|---|---|---|---|---|---|---|
| Community cross-sectional surveys | Zanzibar (multiple) | 2016 | A | 70 | 2–70 | 21 | 52 |
| In vivo efficacy study of artesunate-amodiaquine (ASAQ) with single low-dose primaquine (SLDP) in pediatric uncomplicated malaria patients | Zanzibar (multiple) | 2017 | S | 143 | 2–60 | 117 | 134 |
| Study of transmission of *Plasmodium falciparum* to colony-reared mosquitoes | Mainland Tanzania (Bagamoyo) | 2018 | A | 40 | 7–16 | 34 | 0 |
| Parasite clearance study of artemether-lumefantrine (AL) | Mainland Tanzania (Bagamoyo) | 2018 | S | 138 | 2–11 | 110 | 123 |

*Asymptomatic (A) or symptomatic (S).

samples to assess transmission of parasites within Zanzibar. The larger and spatially rich sample set analyzed in this article offers an opportunity for more refined analyses of transmission across Zanzibar and how parasites are related to those from coastal mainland.

A panel of molecular inversion probes (MIPs), a highly multiplexed genotyping assay, were designed in a previous study to target single-nucleotide polymorphisms (SNPs) throughout the *P. falciparum* genome (*Aydemir et al., 2018*). We leveraged this assay to investigate the genetic epidemiology of parasites in the coastal mainland and Zanzibar utilizing 391 samples collected from cross-sectional surveys of both asymptomatic infections and symptomatic, uncomplicated malaria cases during 2016–2018. Specifically, we use identity by descent (IBD) analyses to compare the genetic relatedness of mainland and Zanzibari parasites, and investigate the geography/spatial relationships of genetically related parasites on the archipelago. We further characterize how the genetic complexity of infections (COIs) differs by clinical status and describe patterns of antimalarial drug resistance polymorphisms in the parasite populations. In this low-transmission setting, these analyses characterize fine-scale local parasite populations that contribute to continued transmission within the region, highlighting a key barrier to malaria elimination in the Zanzibar archipelago.

## Methods

Samples from coastal Tanzania (178) and Zanzibar (213) were previously sequenced through multiple studies (*Table 1*, *Figure 1—figure supplement 1*). These samples include 213 dried blood spots (DBS) collected in Zanzibar between February 2016 and September 2017, coming from cross-sectional surveys of asymptomatic individuals (n = 70) and an in vivo efficacy study of artesunate-amodiaquine (ASAQ) with single low-dose primaquine (SLDP) in pediatric uncomplicated malaria patients in the western and central districts of Unguja island and Micheweni district on Pemba island (n = 143) (*Msellem et al., 2020*). These samples were geolocalized to *shehias*, the lowest geographic governmental designation of land in Zanzibar, across its two main islands, Unguja and the northern region of Pemba (*Figure 1—figure supplement 1*). Mainland Tanzania samples were collected in rural Bagamoyo district, where malaria transmission persists, and residents frequently travel to Dar es Salaam, the major port from where travelers depart for Zanzibar. Of the mainland Bagamoyo samples, 138 were whole blood collected from 2015 to 2017 as part of an in vivo efficacy study of artemether-lumefantrine (AL) in pediatric uncomplicated malaria patients (*Topazian et al., 2022*), and the remaining 40 samples were leukodepleted blood collected in 2018 from asymptomatic but RDT-positive children who participated in a study investigating the transmission of *P. falciparum* to colony-reared mosquitoes. This project leveraged MIP data from SRA, including PRJNA926345, PRJNA454490, PRJNA545345, and PRJNA545347.

In order to place coastal Tanzanian and Zanzibari samples in the context of African *P. falciparum* population structure across multiple regions, MIP data from 147 whole blood samples collected in Ahero District, Kenya, from the same parasite clearance study were used (*Topazian et al., 2022*) in

conjunction with a subset of data from 2537 samples genotyped for a study of the 2013 Demographic Health Survey of the Democratic Republic of the Congo (DRC), which included samples from DRC, Ghana, Tanzania, Uganda, and Zambia (*Verity et al., 2020*; see *Figure 1—figure supplement 2*).

## MIP sequencing

Sequence data for the coastal Tanzanian and Zanzibari samples were generated in a similar fashion across studies. Chelex-extracted DNA from DBS and QIAGEN Miniprep (QIAGEN, Germantown, MD)-extracted DNA from leukodepleted blood were used in MIP captures, which were then sequenced as previously described (*Aydemir et al., 2018*; *Verity et al., 2020*). Control mixtures of four strains of genomic DNA from *P. falciparum* laboratory lines were also sequenced as described previously (*Verity et al., 2020*). We utilized two MIP panels, one being a genome-wide SNP MIP panel and the second being a panel with the known drug resistance markers in *P. falciparum* (*Verity et al., 2020*). These libraries were sequenced on Illumina Nextseq 500 instrument using 150 bp paired end sequencing with dual indexing using Nextseq 500/550 Mid-output Kit v2.

## MIP variant calling and filtering

MIP sequencing data was processed using *MIPTools* (v0.4.0, https://github.com/bailey-lab/MIPTools; *Hathaway, 2024*), which first merges reads and removes errors and unique molecular identifier (UMI) redundancy with *MIPWrangler* (Aydemir, unpublished). For the genome-wide panel, variant calling was performed using *FreeBayes* within *MIPTools*, for a pooled continuous sample that was filtered for a minimum UMI depth of 10, a within-sample allele frequency threshold of 0.01, and a minimum alternate read count of 2 to obtain 5174 variant SNP sites. Utilizing *bcftools* (version 1.15.1), the samples and loci were filtered to only the known targeted SNPs, requiring a minor allele frequency threshold of 0.01, a sample missingness threshold of 10%, and loci missingness threshold of 15%. After filtering and subsetting to biallelic sites, 282 samples were left at 1270 loci. The final numbers of samples used for analysis by group are shown in *Table 1*. Sequencing coverage estimates for loci are shown in *Figure 1—figure supplement 3*.

For the drug resistance panel, variant calling was performed as above, with additional *FreeBayes* parameters of a haplotype length of 3 and using the 30 best alleles at a given locus. Three aggregate amino acid summary tables were created with reference amino acid UMI counts, alternate amino acid UMI counts, and the coverage depth for each variant, for a total of 309 samples at 2265 SNPs. We focused analysis on the following key known and putative drug resistance molecular marker genes and corresponding mutations: *P. falciparum* (*Pf*) chloroquine resistance transporter (Pfcrt: C72S, M74I, N75E, K76T, T93S, H97Y, F145I, I218F, A220S, Q271E, N326S, M343L, C350R, G353V, I356T, R371I), *Pf* multidrug resistance 1 (Pfmdr1: N86Y, Y184F, S1034C, N1042D, D1246Y), *Pf* dihydrofolate reductase (Pfdhfr: A16V, N51I, C59R, S108N, I164L), *Pf* dihydropteroate synthase (Pfdhps: S436A, S436F, A437G, K540E, A581G, A613T, A613S), *Pf* cytochrome b (Pfcytb: Y268N,Y268S,Y268C), and *Pf* kelch 13 (Pfk13: P441L, F446I, G449A, N458Y, C469F, C469Y, M476I, A481V, Y493H, R515K, P527H, N537I, N537D, G538V, R539T, I543T, P553L, R561H, V568G, P574L, C580Y, R622I, A675V) (*World Health Organization, 2020*). Prevalence was calculated separately in Zanzibar or mainland Tanzania for each polymorphism by the number of samples with alternative genotype calls for this polymorphism over the total number of samples genotyped, and an exact 95% confidence interval using the Pearson–Klopper method was calculated for each prevalence.

## Analysis of population relatedness and structure

To investigate genetic relatedness of parasites across regions, IBD estimates were assessed using the within-sample major alleles (coercing samples to monoclonal by calling the dominant allele at each locus) and estimated utilizing a maximum likelihood approach using the *inbreeding_mle* function from the *MIPanalyzer* package (*Verity et al., 2020*). This approach has previously been validated as a conservative estimate of IBD (*Verity et al., 2020*). Next, principal component analysis (PCA) was performed to query the comparative genetic variation of the samples by utilizing the genome-wide SNP panel. We pruned 51 samples that had a pairwise IBD of >0.90 to one randomly selected sample as a representative of the clonal population to avoid clonal structure from dominating the analysis. Within-sample allele frequencies were calculated, with an imputation step replacing missing values

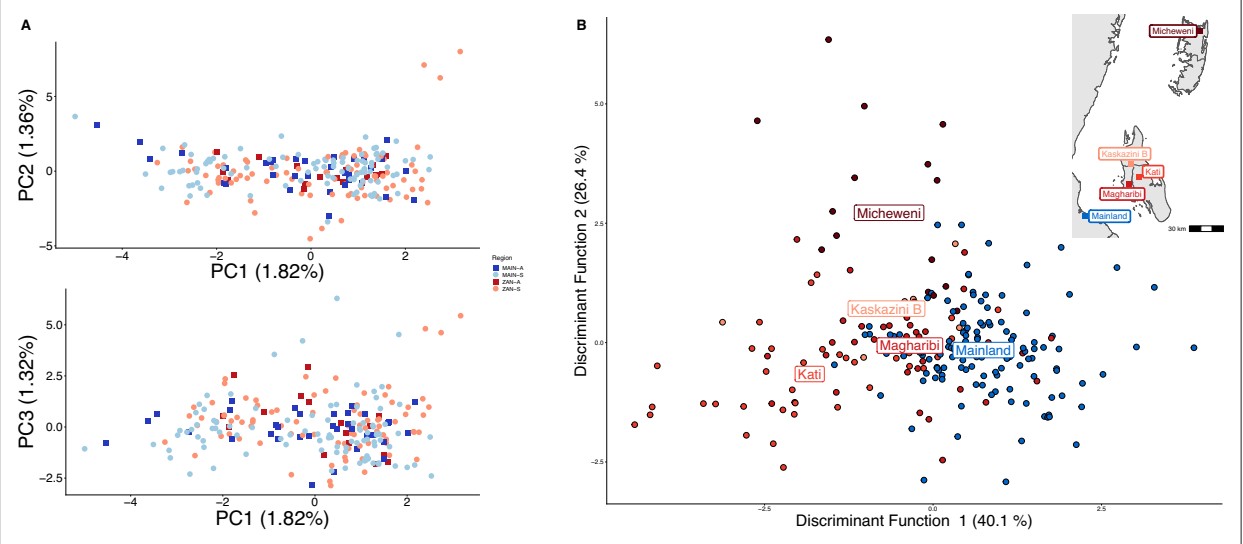

**Figure 1.** Parasites between Zanzibar and coastal mainland Tanzania are highly related but microstructure within Zanzibar is apparent. (**A**) Principal component analysis (PCA) comparing parasites from symptomatic vs. asymptomatic patients from coastal Tanzania and Zanzibar. Clusters with an identity by descent (IBD) value of > 0.90 were limited to a single representative infection to prevent local structure of highly related isolates within *shehias* from driving clustering. (**B**) A discriminant analysis of principal components (DAPC) was performed utilizing isolates with unique pseudohaplotypes, pruning highly related isolates to a single representative infection. Districts were included with at least five isolates remaining to have sufficient samples for the DAPC. For plotting the inset map, the district coordinates (e.g., mainland, Kati, etc.) were calculated from the averages of the *shehia* centroids within each district.

The online version of this article includes the following figure supplement(s) for figure 1:

**Figure supplement 1.** Sampling locations in Zanzibar (*shehia*) and mainland (Bagamoyo district) Tanzania.

**Figure supplement 2.** Principal component analysis (PCA) utilizing samples across Africa shows clustering based on geographic location.

**Figure supplement 3.** Molecular inversion probe (MIP) performance shows coverage of loci.

**Figure supplement 4.** Principal component analysis (PCA) with highly related samples shows population stratification radiating from coastal mainland to Zanzibar.

with the median per each locus, and PCA was performed using the *prcomp* function (***Verity et al., 2020***; *R* version 4.2.1).

To include geographic information with querying genetic variation, discriminant analysis of principal components (DAPC) was used (***Jombart, 2008***). Pseudohaplotypes were created by pruning the genotype calls at all loci for each sample into a single haplotype, and redundant haplotypes were removed (282 reduced to 272 with unique pseudohaplotypes). DAPC was conducted at the district level, and samples from districts with less than five samples (272 samples to 270 samples in six districts) were retained (***Figure 1—figure supplement 4B***). For the main DAPC (***Figure 1B***), highly related isolates were pruned to a single representative infection (272 reduced to 232) and then included districts with at least five samples (232 reduced to 228 samples in five districts). The DAPC was performed using the *adegenet* package (***Jombart, 2008***) with the first 80 PCs based on the cross-validation function *xvalDapc*. To perform K-means clustering, a cluster K of 1 was assigned to the mainland samples while the *kmeans* package was used to find the optimal K to cluster the Zanzibar *shehias* by latitude and longitude. The K-means clustering experiment was used to cluster a continuous space of geographic coordinates in order to compare genetic relatedness in different regions. We selected K = 4 as the inflection point based on the elbow plot (***Figure 2—figure supplement 1***) and based the number to obtain sufficient subsections of Zanzibar to compare genetic relatedness.

To investigate how genetic relatedness varies as the distance between pairs increases, isolation by distance was performed across all of Zanzibar and within the islands of Unguja and Pemba. The greater circle (GC) distances between each *shehia* centroid were calculated (within *shehia* distances were equal to 0) to find the distance between each *shehia* in geographic space. Distances were then binned at increments reflecting the max GC distances between regions, which was smallest in Pemba at 12 km and much larger for both Unguja (58 km) and all of Zanzibar (135 km). Within each binned

group, the mean IBD with 95% CIs was plotted. For graphing IBD connections at the between and within *shehia* level, an IBD threshold of 0.25 (half-siblings) or greater was used (*Figure 4—figure supplements 1 and 2*). In graphing IBD connections at larger distances between islands or between coastal mainland Tanzania and Zanzibar, a between IBD value of 0.125 (quarter-siblings) or greater was used (*Figure 4—figure supplements 3 and 4*). These plots were created utilizing *ggraph* in *R* with the nodes being samples and the edges being IBD estimates.

COI, or the number of parasite clones in a given sample, was determined using *THE REAL McCOIL* (v2) categorical method (*Chang et al., 2017*) and the 95% CI was calculated utilizing a nonparametric bootstrap. Fws statistic, which is used to compare the diversity within and between samples in a population, was calculated in *R* version 4.2.1 through the formula, $(1 - H_w)/H_p$, where $H_w$ is the within-sample heterozygosity and $H_p$ is the heterozygosity across the population, and 95% CIs were calculated utilizing a nonparametric bootstrap.

## Results

### Zanzibari falciparum parasites were closely related to coastal mainland parasites but showed higher within- than between-population IBD and evidence of microstructure on the archipelago

To examine geographic relatedness, we first used PCA. Zanzibari parasites are highly related to other parasites from East Africa and more distantly related to Central and West African isolates (*Figure 1—figure supplement 2*). PCA of 232 coastal Tanzanian and Zanzibari isolates, after pruning 51 samples with an IBD of >0.9 to one representative sample, demonstrates little population differentiation (*Figure 1A*).

However, after performing K-means clustering of *shehias* in Zanzibar and mainland Tanzania, parasites within each population show more highly related pairs within their respective clusters than between clusters (*Figure 2*). Comparisons of parasite pairs between Zanzibar and coastal Tanzania showed no pairs with an IBD >0.20 (*Figure 2*, *Figure 4—figure supplement 3*). Similarly, no pairs with an IBD of ≥0.20 were present in pairwise comparisons between Unguja and Pemba (*Figure 4—figure supplement 4*).

To further assess the differentiation within the parasite population in Zanzibar, we conducted DAPC according to the districts of origin for each isolate. Parasites differentiated geographically, with less variation near the port of Zanzibar town and more differentiation in isolates collected in districts further from the port (*Figure 1B*). This underlying microstructure is also supported by classic isolation by distance analysis (*Figure 3*, *Figure 3—figure supplement 1*). Isolation by distance analysis across all of Zanzibar and within Unguja showed rapid decay of relatedness over very short geographic distances (*Figure 3A and B*). Interestingly in Pemba, mean IBD remained at a similar relatively high level even at longer distances (*Figure 3C*).

### Within Zanzibar, parasite clones are shared within and between *shehias*, suggesting local outbreaks

Among the sample pairs in Zanzibar that are highly related (IBD of ≥0.25), we see different patterns of genetic relatedness suggesting common local and short-distance transmission of clones and occasional long-distance transmission (*Figure 4*). In Unguja (*Figure 4A*), we see multiple identical or near-identical parasite pairs shared over longer distances, suggesting longer distance gene flow, as well as multiple *shehias* containing highly related pairs. In northern Pemba, there is one large cluster of highly related parasites shared within and between six *shehias* (*Figure 4B*). Network analysis (*Figure 4C*) for all sample pairs with an IBD of >0.25 from these *shehias* illustrates this, with pairs linked by yellow lines showing the highest IBD. The largest network represents two highly related clusters (groups linked by yellow edges, mean IBD of 0.99) connected by a highly related intermediate (FMH42), suggesting that the clusters are related through parasites that have recombined while on the archipelago. FMH42 links the lower cluster with pairwise IBD of 0.65 and the upper cluster with a pairwise IBD of 0.27. These symptomatic isolates collected from February 2016 to September 2017 in northern Pemba likely derive from sustained transmission from a seeding event.

Network analysis of within *shehia* pairwise IBD sharing in Unguja again shows that there is close relatedness on this small geographic scale (*Figure 4—figure supplement 1*). A cluster of four isolates

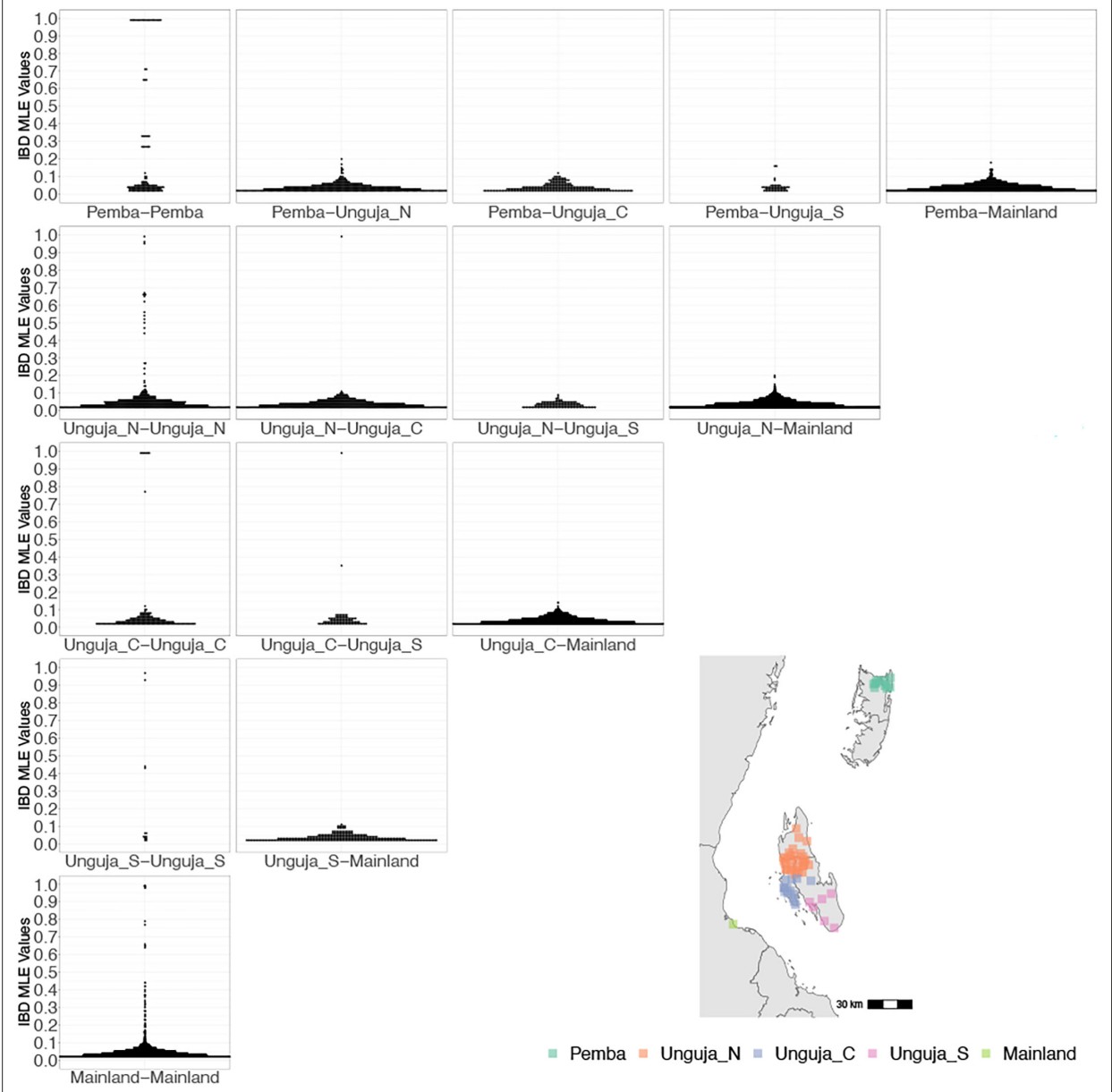

**Figure 2.** Coastal Tanzania and Zanzibari parasites have more highly related pairs within their given region than between regions. K-means clustering of *shehia* coordinates was performed using geographic coordinates of all *shehias* present from the sample population to generate five clusters (colored boxes). All *shehias* were included to assay pairwise identity by descent (IBD) between differences throughout Zanzibar. K-means cluster assignments were converted into interpretable geographic names Pemba, Unguja North (Unguja_N), Unguja Central (Unguja_C), Unguja South (Unguja_S), and mainland Tanzania (Mainland). Pairwise comparisons of within-cluster IBD (column 1 of IBD distribution plots) and between-cluster IBD (columns 2–5 of IBD distribution plots) were done for all clusters. All IBD values > 0 were plotted for each comparison. In general, within-cluster IBD had more pairwise comparisons containing high IBD identity.

The online version of this article includes the following figure supplement(s) for figure 2:

**Figure supplement 1.** Diagnostic plot showing total within-cluster sum of squares versus number of clusters for the determination of optimal K.

in the Shakani *shehia* on Unguja island with pairwise IBDs of 0.99 likely reflects ongoing transmission within Shakani, with similar connections in Bambi and Dimani. Meanwhile, a few distant connections likely reflect the extent of human mobility on the island (*Figure 4A*). Similar within-district networks on the mainland are shown in *Figure 4—figure supplement 2*.

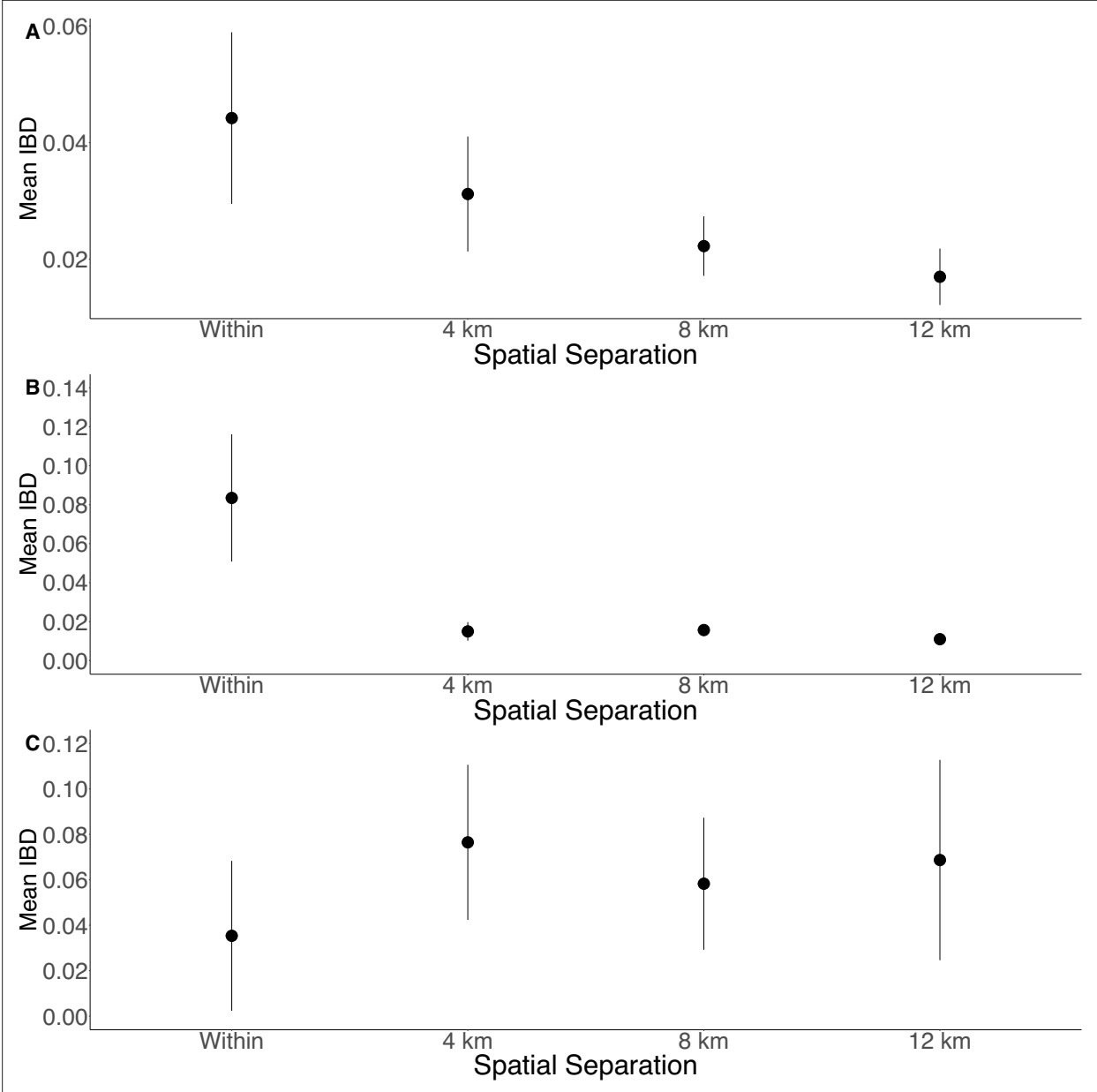

**Figure 3.** Isolation by distance is shown between all Zanzibari parasites (**A**), only Unguja parasites (**B**), and only Pemba parasites (**C**). Samples were analyzed based on geographic location. Zanzibar (N = 136) (**A**), Unguja (N = 105) (**B**), or Pemba (N = 31) (**C**) and greater circle (GC) distances between pairs of parasite isolates were calculated based on *shehia* centroid coordinates. These distances were binned at 4 km increments out to 12 km. Identity by descent (IBD) beyond 12 km is shown in *Figure 3—figure supplement 1*. The maximum GC distance for all of Zanzibar was 135 km, 58 km on Unguja, and 12 km on Pemba. The mean IBD and 95% CI are plotted for each bin.

The online version of this article includes the following figure supplement(s) for figure 3:

**Figure supplement 1.** Isolation by distance in Zanzibar isolates (**A**) and only Unguja isolates (**B**).

## Compared to symptomatic infections, asymptomatic infections demonstrate greater genetic complexity, especially in coastal Tanzania

Asymptomatic infections were compared to roughly contemporaneously collected isolates from those presenting with acute, uncomplicated malaria. Asymptomatic infections demonstrated greater COI than symptomatic infections on both the coastal mainland (mean COI 2.5 vs 1.7, p<0.05, Wilcoxon–Mann–Whitney test) and in Zanzibar (mean COI 2.2 vs 1.7, p=0.05, Wilcoxon–Mann–Whitney test) (*Figure 5A*). A similar pattern was seen when evaluating Fws, which measures the diversity within a

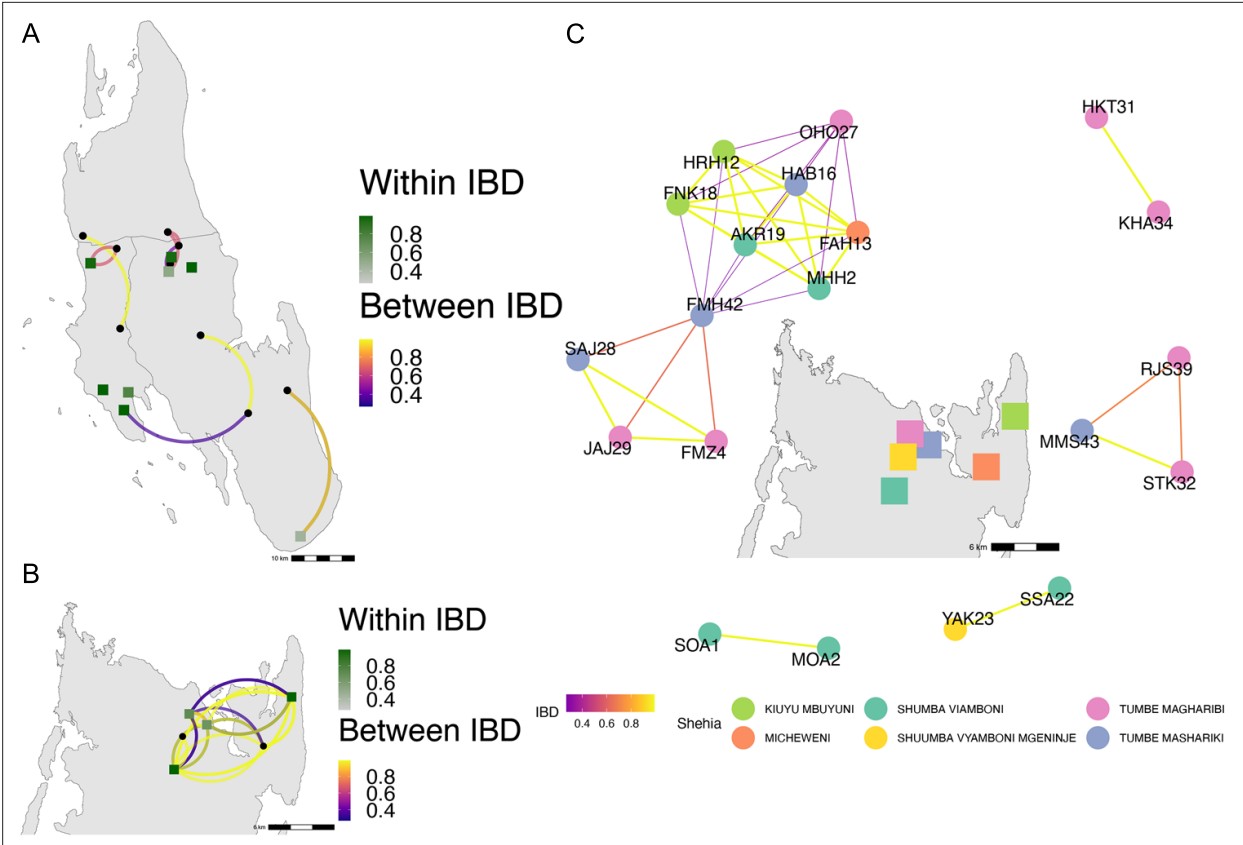

**Figure 4.** Highly related pairs span long distances across Zanzibar. Sample pairs were filtered to have identity by descent (IBD) estimates of ≥ 0.25. Within *shehia* pairwise IBD estimates are shown within Unguja (**A**) and Pemba (**B**) as single points, with dark green representing the greatest degree of IBD. *Shehias* labeled with black dots do not have within IBD estimates of ≥ 0.25. Between *shehia* IBD reflects pairs of parasites with IBD ≥0.25, with the color of the connecting arc representing the degree of IBD and yellow representing maximal connectivity. Panel (**C**) shows the network of highly related pairs (IBD ≥ 0.25) within and between the six northern Pemba *shehias* (note: Micheweni is a *shehia* in Micheweni district). Samples (nodes) are colored by *shehia* and IBD estimates (edges) are represented on a continuous scale with increasing width and yellow-shading indicating higher IBD.

The online version of this article includes the following figure supplement(s) for figure 4:

**Figure supplement 1.** Network analysis of within *shehia* comparisons with an identity by descent (IBD) of ≥0.25 in Unguja.

**Figure supplement 2.** Network analysis of sample pairs with an identity by descent (IBD) of ≥0.25 for coastal mainland Tanzania.

**Figure supplement 3.** Sample pairs with an identity by descent (IBD) of ≥0.125 between Zanzibar and mainland Tanzania.

**Figure supplement 4.** Sample pairs with an identity by descent (IBD) of ≥0.125 between Unguja and Pemba.

sample compared to the population, with lower Fws in asymptomatic samples consistent with higher within-host complexity, with a more pronounced difference on the mainland (*Figure 5B*). Despite these differences, parasites from asymptomatic and symptomatic infections tended to cluster together in PCA, suggesting that their core genomes are genetically similar and do not vary based on clinical status (*Figure 1A*).

## Drug resistance mutations did not vary between populations

The prevalence of the drug resistance genotypes was quite similar in Zanzibar and coastal Tanzania (*Table 2*). The frequencies of five mutations associated with sulfadoxine/pyrimethamine resistance (Pfdhfr: N51I, C59R, S108N, Pfdhps: A437G, K540E) were quite high, with prevalences at or above 0.90. Pfcrt mutations associated with chloroquine and amodiaquine resistance (M74I, N75E, K76T) were all present at approximately 0.05 prevalence (*Djimdé et al., 2001*; *Holmgren et al., 2006*). For Pfmdr1, wild-type N86 and D1246 were dominant at 0.99 prevalence, which are associated with reduced susceptibility to lumefantrine (*Sisowath et al., 2005*). No World Health Organization-validated or candidate polymorphism in Pfk13 associated with artemisinin resistance was found.

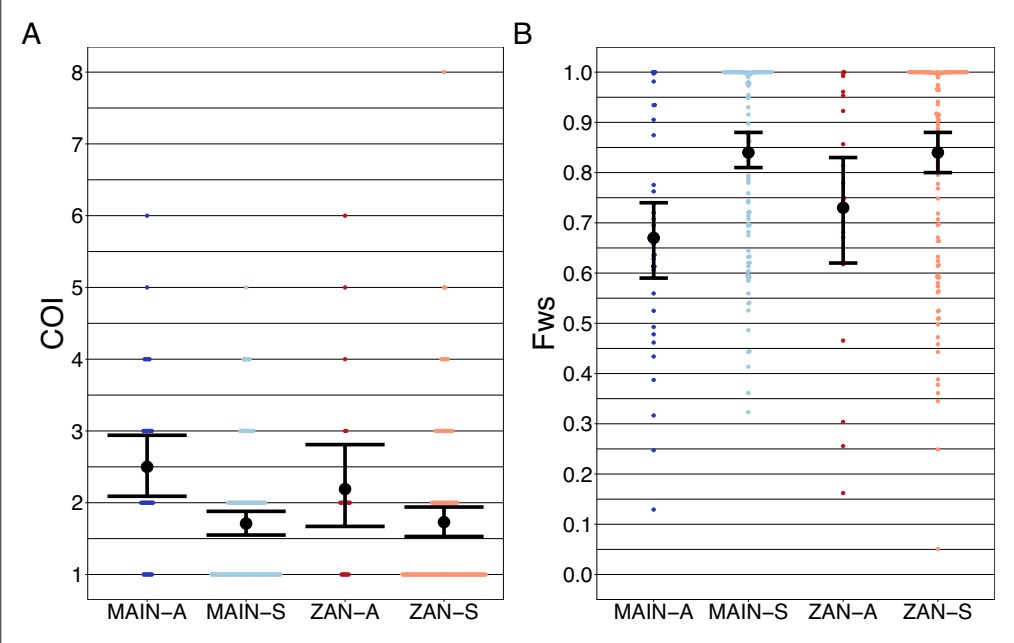

**Figure 5.** Complexity of infection (COI) and Fws metric shows a higher COI and lower Fws in asymptomatic than symptomatic infections in both mainland Tanzania and Zanzibar isolates. COI (**A**) was estimated using the REAL McCOIL's categorical method (*Chang et al., 2017*). The mean COI for asymptomatic was greater than symptomatic infections for all regions; MAIN-A: 2.5 (2.1–2.9), MAIN-S: 1.7 (1.6–1.9), p<0.05, Wilcoxon–Mann–Whitney test and ZAN-A: 2.2 (1.7–2.8), ZAN-S: 1.7 (1.5–1.9), p=0.05, Wilcoxon–Mann–Whitney test. Fws (**B**) was estimated utilizing the formula, $(1-H_w)/H_p$, where $H_w$ is the within-sample heterozygosity and $H_p$ is the heterozygosity across the population. Mean Fws was less in asymptomatic than symptomatic samples; MAIN-A: 0.67 (0.6–0.7), MAIN-S: 0.85 (0.8–0.9), p<0.05, Wilcoxon–Mann–Whitney test and ZAN-A: 0.73 (0.6–0.8), ZAN-S: 0.84 (0.8–0.9), p=0.05, Wilcoxon–Mann–Whitney test. A nonparametric bootstrap was applied to calculate the mean and 95% CI from the COI and Fws values.

## Discussion

In this study, we leverage high-throughput targeted sequencing using MIPs to characterize the populations and the relationships of *P. falciparum* isolates in Zanzibar and coastal mainland Tanzania. The parasite populations appear to be highly related to each other (*Figure 1A*) when evaluated using SNPs in the core genome. Interestingly, within Zanzibar, structure could be observed, with parasites closer to the main ferry terminal in Zanzibar town clustered more closely with coastal mainland parasites (*Figure 1B*, *Figure 1—figure supplement 1*) compared to parasites that were more geographically distant. This, in combination with the evidence of rapid decline of genetic relatedness with distance on the archipelago (*Figure 3*), is consistent with population microstructure within the island chain. This microstructure within the archipelago is supported by K-means clustering where Zanzibari isolates show higher within-cluster than between-cluster IBD (*Figure 2*). It is also consistent with isolates with higher IBD (*Figure 4A and B*) in Unguja and Pemba compared to a maximum IBD of 0.20 between Zanzibar and coastal mainland Tanzania (*Figure 4—figure supplement 3*) or between Unguja and Pemba (*Figure 4—figure supplement 4*). Parasite populations within the very low transmission region of Zanzibar may be more isolated than expected, allowing them to differentiate from each other. This may be indicative of very effective local malaria control, yet with continued microtransmission remaining. Thus, directly targeting local malaria transmission, including the asymptomatic reservoir which contributes to sustained transmission (*Barry et al., 2021*; *Sumner et al., 2021*), may be an important focus for ultimately achieving malaria control in the archipelago (*Björkman and Morris, 2020*). Currently, a reactive case detection program within index case households is being implemented, but local transmission continues and further investigation into how best to control this is warranted (*Mkali et al., 2023*).

**Table 2.** Drug resistance polymorphism prevalence in Zanzibar and coastal mainland Tanzania.

| | Zanzibar | | | Mainland Tanzania | | |
|---|---|---|---|---|---|---|
| Mutation | Mutant allele prevalence* | CI[†] | # Genotyped samples [‡] | Mutant allele prevalence* | CI[†] | # Genotyped samples [‡] |
| Pfcrt-M74I | 0.054 | 0.026–0.098 | 184 | 0.000 | 0–0.034 | 106 |
| Pfcrt-N75E | 0.054 | 0.026–0.098 | 184 | 0.000 | 0–0.034 | 106 |
| Pfcrt-K76T | 0.054 | 0.026–0.098 | 184 | 0.000 | 0–0.034 | 106 |
| Pfdhfr-A16V | 0.000 | 0–0.021 | 173 | 0.000 | 0–0.032 | 112 |
| Pfdhfr-N51I | 0.977 | 0.943–0.994 | 177 | 0.964 | 0.911–0.99 | 112 |
| Pfdhfr-C59R | 0.971 | 0.934–0.991 | 174 | 0.945 | 0.884–0.98 | 109 |
| Pfdhfr-S108N | 1.000 | 0.98–1 | 179 | 1.000 | 0.965–1 | 104 |
| Pfdhfr-S108T | 0.000 | 0–0.02 | 179 | 0.000 | 0–0.035 | 104 |
| Pfdhfr-I164L | 0.000 | 0–0.02 | 184 | 0.000 | 0–0.037 | 98 |
| Pfdhps-A437G | 1.000 | 0.98–1 | 182 | 1.000 | 0.968–1 | 115 |
| Pfdhps-K540E | 0.955 | 0.913–0.98 | 178 | 0.964 | 0.91–0.99 | 111 |
| Pfdhps-A581G | 0.044 | 0.019–0.085 | 181 | 0.107 | 0.058–0.175 | 122 |
| Pfk13-K189N | 0.023 | 0.006–0.058 | 174 | 0.000 | 0–0.04 | 90 |
| Pfk13-K189T | 0.078 | 0.042–0.13 | 166 | 0.095 | 0.042–0.179 | 84 |
| Pfmdr1-N86Y | 0.011 | 0.001–0.04 | 180 | 0.008 | 0–0.044 | 124 |
| Pfmdr1-Y184F | 0.644 | 0.57–0.714 | 180 | 0.530 | 0.435–0.624 | 115 |
| Pfmdr1-D1246Y | 0.011 | 0.001–0.039 | 184 | 0.019 | 0.002–0.067 | 105 |
| Pfmdr2-I492V | 0.430 | 0.357–0.506 | 179 | 0.407 | 0.302–0.518 | 86 |

*Prevalence was calculated as described in the 'Methods'.
[†]95% CI of these polymorphisms were calculated using the Pearson–Klopper method.
[‡]The number of genotyped samples per loci is also shown for each polymorphism.

Despite the overall genetic similarity between archipelago populations, we did not find parasite pairs with high levels of IBD between the coastal mainland and Zanzibar, with the highest being 0.20. While this level still represents a significant amount of genetic sharing, similar to a cousin, the lack of higher levels does not allow us to identify specific importation events. This is largely due to the study design, which is based on convenience sampling, the relatively low numbers of samples, and lack of sampling from all mainland travel hubs (*Bisanzio et al., 2023*). Sampling was also denser in Unguja compared to Pemba. On the other hand, we see clear transmission of highly related parasites within each population (IBD > 0.99). In Zanzibar, highly related parasites mainly occur in the range of 20–30 km. These results are similar to our previous work using whole-genome sequencing of isolates from Zanzibar and mainland Tanzania, showing increased within-population IBD compared to between-population IBD (*Morgan et al., 2020*). The network of highly related *P. falciparum* parasites from six *shehias* in north Pemba provides an excellent example of likely recent near-clonal transmission, consistent with an outbreak (*Figure 4C*). A recent study investigating population structure in Zanzibar also found local population microstructure in Pemba (*Holzschuh et al., 2023*). Furthermore, both studies found near-clonal parasites within the same district, Micheweni, and found population microstructure over Zanzibar. Overall, given the findings of microstructure with significant local sharing of highly related strains, these small clusters still potentially drive much of the malaria transmission that occurs within the archipelago through routine human movement or mosquito travel between locales (*Huestis et al., 2019*). Less frequent longer distance transmission events also occur, likely due to longer range human migration within the islands.

Asymptomatic parasitemia has been shown to be common in falciparum malaria around the globe and has been shown to have increasing importance in Zanzibar (*Lindblade et al., 2013*;

*Morris et al., 2015*). What underlies the biology and prevalence of asymptomatic parasitemia in very low transmission settings where antiparasite immunity is not expected to be prevalent remains unclear (*Björkman and Morris, 2020*). Similar to a few previous studies, we found that asymptomatic infections had a higher COI than symptomatic infections across both the coastal mainland and Zanzibar parasite populations (*Collins et al., 2022*; *Kimenyi et al., 2022*; *Sarah-Matio et al., 2022*). Other studies have found lower COI in severe vs. mild malaria cases (*Robert et al., 1996*) or no significant difference between COI based on clinical status (*Conway et al., 1991*; *Earland et al., 2019*; *Kun et al., 1998*; *Lagnika et al., 2022*; *Tanabe et al., 2015*). In Zambia, one study suggested that infections that cause asymptomatic infection may be genetically different from those that cause symptomatic infection (*Searle et al., 2017*). However, this study included samples collected over different time periods and relied on a low-density genotyping assay that only investigated the diversity of 24 SNPs across the genome. Here, based on SNPs throughout the core genome, we did not see differential clustering of asymptomatic or symptomatic infections in Zanzibar or the mainland (*Figure 1A*), suggesting that these parasite populations remain similar when comparing clinical status. However, this genotyping approach does not address potential variation in the many hypervariable gene families that encode genes known to be associated with pathogenesis (e.g., *var*, *rifin,* and *stevor* genes) and does not address differences in expression of genes associated with pathogenesis that may reflect differences in the populations. Investigation with other methods, such as long-read genome sequencing and transcriptional profiling, would be needed to address these differences.

While mutations for partial artemisinin resistance were not observed in K13, other antimalarial-resistant mutations of concern were observed. Validated drug resistance mutations linked to sulfadoxine/pyrimethamine resistance (Pfdhfr-N51I, Pfdhfr-C59R, Pfdhfr-S108N, Pfdhps-A437G, Pfdhps-K540E) were found at high prevalence (*Table 2*). Prevalence of polymorphisms associated with amodiaquine resistance (Pfcrt-K76T, Pfmdr1-N86Y, Pfmdr1-Y184F, Pfmdr1-D1246Y) was seen at similar proportions as previous reports (*Msellem et al., 2020*). The wild-type Pfmdr1-N86 was dominant in both mainland and archipelago populations, concerning reduced lumefantrine susceptibility. Although polymorphisms associated with artemisinin resistance did not appear in this population, continued surveillance is warranted given the emergence of these mutations in East Africa and reports of rare resistance mutations on the coast consistent with the spread of emerging Pfk13 mutations (*Moser et al., 2021*).

Overall, parasites between Zanzibar and coastal mainland Tanzania remain highly related, but population microstructure on the island reflects ongoing low-level transmission in Zanzibar, partially driven by asymptomatic infections that potentially constitute a long-term reservoir. This is likely the result of the continued pressure on the population through the implementation of effective control measures. In this study, parasite genomics allows us to parse differences in parasite populations and reveals substructure in an area of low-transmission intensity. A recent study identified 'hotspot' *shehias*, defined as areas with comparatively higher malaria transmission than other *shehias*, near the port of Zanzibar town and in northern Pemba (*Bisanzio et al., 2023*). These regions overlapped with *shehias* in this study with high levels of IBD, especially in northern Pemba (*Figure 4*). These areas of substructure represent parasites that differentiated in relative isolation and are thus important locales to target intervention to interrupt local transmission (*Bousema et al., 2012*). While a field cluster-randomized control trial in Kenya targeting these hotspots did not confer much reduction of malaria outside of the hotspot (*Bousema et al., 2016*), if areas are isolated pockets, which genetic differentiation can help determine, targeted interventions in these areas are likely needed, potentially through both mass drug administration and vector control (*Morris et al., 2018*; *Okell et al., 2011*). Such strategies and measures preventing imported malaria could accelerate progress toward zero malaria in Zanzibar.

## Acknowledgements

We thank the communities and participants who took part in these studies. We also thank Abebe Fola for his assistance. We acknowledge the following institutional funding - MRC Centre for Global Infectious Disease Analysis, jointly funded by the UK MRC and the UK FCDO, under the MRC/FCDO Concordat agreement and is also part of the EDCTP2 programme supported by the EU.

# Additional information

## Funding

| Funder | Grant reference number | Author |
|---|---|---|
| National Institutes of Health | R01AI121558 | Jonathan J Juliano<br>Jeffrey A Bailey |
| National Institutes of Health | R01AI137395 | Billy E Ngasala<br>Zackary Park<br>Lwidiko E Mhamilawa<br>Jessica T Lin<br>Jonathan J Juliano<br>Jeffrey A Bailey |
| National Institutes of Health | R01AI155730 | Sean V Connelly<br>Billy E Ngasala<br>Varun Goel<br>Robert Verity<br>Jessica T Lin<br>Anders Björkman<br>Jonathan J Juliano<br>Jeffrey A Bailey |
| National Institutes of Health | F30AI143172 | Nicholas F Brazeau |
| National Institutes of Health | K24AI134990 | Jonathan J Juliano |
| Swedish Research Council | | Ulrika Morris<br>Andreas Mårtensson<br>Anders Björkman |
| Erling-Persson Family Foundation | | Anders Björkman |
| Yang Biomedical Scholars Fund | | Jonathan J Juliano |
| Community Jameel | | Robert Verity |

The funders had no role in study design, data collection and interpretation, or the decision to submit the work for publication.

## Author contributions

Sean V Connelly, Conceptualization, Formal analysis, Validation, Investigation, Visualization, Methodology, Writing – original draft, Writing – review and editing; Nicholas F Brazeau, Conceptualization, Formal analysis, Methodology, Writing – review and editing; Mwinyi Msellem, Resources, Data curation, Writing – review and editing; Billy E Ngasala, John M Ong'echa, Conceptualization, Resources, Funding acquisition, Writing – review and editing; Ozkan Aydemir, Conceptualization, Data curation, Formal analysis, Writing – review and editing; Varun Goel, Karamoko Niaré, Formal analysis, Writing – review and editing; David J Giesbrecht, Zachary R Popkin-Hall, Chris Hennelly, Zackary Park, Investigation, Writing – review and editing; Ann M Moormann, Conceptualization, Funding acquisition, Writing – review and editing; Robert Verity, Conceptualization, Methodology, Writing – review and editing; Safia Mohammed, Shija J Shija, Resources, Writing – review and editing; Lwidiko E Mhamilawa, Conceptualization, Resources, Data curation, Writing – review and editing; Ulrika Morris, Conceptualization, Resources, Writing – review and editing; Andreas Mårtensson, Conceptualization, Resources, Data curation, Supervision, Investigation, Writing – original draft, Project administration, Writing – review and editing; Jessica T Lin, Conceptualization, Supervision, Funding acquisition, Investigation, Writing – original draft, Project administration, Writing – review and editing; Anders Björkman, Conceptualization, Resources, Supervision, Funding acquisition, Investigation, Writing – original draft, Project administration, Writing – review and editing; Jonathan J Juliano, Jeffrey A Bailey, Conceptualization, Resources, Data curation, Supervision, Funding acquisition, Investigation, Writing – original draft, Project administration, Writing – review and editing

## Author ORCIDs

Sean V Connelly ⓘ http://orcid.org/0000-0002-7330-7340
Ozkan Aydemir ⓘ http://orcid.org/0000-0002-3458-6527
John M Ong'echa ⓘ http://orcid.org/0000-0003-3928-6774
Ulrika Morris ⓘ http://orcid.org/0000-0003-4734-5754
Jessica T Lin ⓘ https://orcid.org/0000-0002-4516-723X
Jonathan J Juliano ⓘ http://orcid.org/0000-0002-0591-0850
Jeffrey A Bailey ⓘ https://orcid.org/0000-0002-6899-8204

## Ethics

All participants provided informed consent for analysis and publication. The IRBs of University of North Carolina at Chapel Hill (15-1989, 17-0166, 18-1090), Muhimbili University of Health and Allied Sciences, Zanzibar Medical Research Ethical Committee and the Regional Ethics Review Board, Stockholm, Sweden gave ethical approval for this work.

Reviewer #1 (Public Review): https://doi.org/10.7554/eLife.90173.3.sa1
Reviewer #2 (Public Review): https://doi.org/10.7554/eLife.90173.3.sa2
Author response https://doi.org/10.7554/eLife.90173.3.sa3

# Additional files

## Supplementary files

• MDAR checklist

## Data availability

Parasite sequence data is available through SRA (BioProject PRJNA926345). Code used for analysis is available at: https://github.com/sconnelly007/TAN_MIP (copy archived at *Connelly, 2024*).

The following dataset was generated:

| Author(s) | Year | Dataset title | Dataset URL | Database and Identifier |
|---|---|---|---|---|
| Connelly SV, Brazeau NF, Msellem M, Ngasala BE, Aydemir O, Goel V, Niaré K, Giesbrecht DJ, Popkin-Hall ZR, Hennelly CM, Park Z, Moormann AM, Ong'echa JM, Verity R, Mohammed S, Shija SJ, Mhamilawa LE, Morris U, Mårtensson A, Lin JT, Björkman A, Juliano JJ, Bailey JA | 2023 | Strong isolation by distance and evidence of population microstructure reflect ongoing *Plasmodium falciparum* transmission in Zanzibar | https://www.ncbi.nlm.nih.gov/bioproject/?term=PRJNA926345 | NCBI BioProject, PRJNA926345 |

The following previously published datasets were used:

| Author(s) | Year | Dataset title | Dataset URL | Database and Identifier |
|---|---|---|---|---|
| Verity R, Aydemir O, Brazeau NF, Watson OJ, Hathaway NJ, Mwandagalirwa MK, Marsh PW, Thwai K, Fulton T, Denton M, Morgan AP, Parr JB, Tumwebaze PK, Conrad M, Rosenthal PJ, Ishengoma DS, Ngondi J, Gutman J, Mulenga M, Norris DE, Moss WJ, Mensah BA, Myers-Hansen JL, Ghansah A, Tshefu AK, Ghani AC, Meshnick SR, Bailey JA, Juliano JJ | 2018 | *Plasmodium falciparum* (malaria parasite *P. falciparum*) | https://www.ncbi.nlm.nih.gov/bioproject/?term=PRJNA454490 | NCBI BioProject, PRJNA454490 |
| Verity R, Aydemir O, Brazeau NF, Watson OJ, Hathaway NJ, Mwandagalirwa MK, Marsh PW, Thwai K, Fulton T, Denton M, Morgan AP, Parr JB, Tumwebaze PK, Conrad M, Rosenthal PJ, Ishengoma DS, Ngondi J, Gutman J, Mulenga M, Norris DE, Moss WJ, Mensah BA, Myers-Hansen JL, Ghansah A, Tshefu AK, Ghani AC, Meshnick SR, Bailey JA, Juliano JJ | 2019 | *Plasmodium falciparum* (malaria parasite *P. falciparum*) | https://www.ncbi.nlm.nih.gov/bioproject/?term=PRJNA545345 | NCBI BioProject, PRJNA545345 |
| Verity R, Aydemir O, Brazeau NF, Watson OJ, Hathaway NJ, Mwandagalirwa MK, Marsh PW, Thwai K, Fulton T, Denton M, Morgan AP, Parr JB, Tumwebaze PK, Conrad M, Rosenthal PJ, Ishengoma DS, Ngondi J, Gutman J, Mulenga M, Norris DE, Moss WJ, Mensah BA, Myers-Hansen JL, Ghansah A, Tshefu AK, Ghani AC, Meshnick SR, Bailey JA, Juliano JJ | 2020 | *Plasmodium falciparum* (malaria parasite *P. falciparum*) | https://www.ncbi.nlm.nih.gov/bioproject/?term=PRJNA545347 | NCBI BioProject, PRJNA545347 |

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
