## [Editor Report · eLife assessment]

Connelly and colleagues provide **convincing** genetic evidence that importation from mainland Tanzania is a major source of *Plasmodium falciparum* lineages currently circulating in Zanzibar. This study also reveals ongoing local malaria transmission and occasional near-clonal outbreaks in Zanzibar. Overall, the article effectively highlights the role of human movements in maintaining residual malaria transmission in an area targeted for intensive control interventions over the past decades and provides clear and **valuable** information for epidemiologists and public health professionals.

---

## [Referee Report · Reviewer #1 (Public Review)]

Summary:

Zanzibar archipelago is close to achieve malaria elimination, but despite the implementation of effective control measures there is still low level seasonal malaria transmission. This could be due to the frequent importation of malaria from the mainland Tanzania and Kenya, reservoir of asymptomatic infections and competent vectors. To investigate population structure and gene flow of *P. falciparum* in Zanzibar and mainland Tanzania, they used 178 samples from mainland Tanzania and 213 from Zanzibar that were previously sequenced using molecular inversion probes (MIPs) panels targeting single nucleotide polymorphisms (SNPs). They performed Principal Component Analysis (PCA) and identity by descent (IBD) analysis to assess genetic reladness between isolates. Parasites from coastal mainland Tanzania contribute for the genetic diversity in parasite population in Zanzibar. Despite this, there is a pattern of isolation by distance and microstructure within the achipelago, and evidence of local sharing of highly related strains sustaining malaria transmission in Zanzibar that are important targets for interventions such as mass drug administration and vector control, in addition to measures against imported malaria.

Strengths:

This study presents important samples to understand population structure and gene flow between mainland Tanzania and Zanzibar, especially from rural Bagamoyo District, where malaria transmission persists and there is a major port of entry to Zanzibar. In addition, this study includes a larger set of SNPs, providing more robustness for analyzes such as PCA and IBD. Therefore, the conclusions of this paper are well supported by data.

Comments on revised version:

The authors answered all my questions.

---

## [Referee Report · Reviewer #2 (Public Review)]

Summary:

This manuscript describes *P. falciparum* population structure in Zanzibar and mainland Tanzania. 282 samples were typed using molecular inversion probes. The manuscript is overall well written and shows clear population structure. It follows a similar manuscript published earlier this year, which typed a similar number of samples collected mostly in the same sites around the same time. The current manuscript extends this work by including a large number of samples from coastal Tanzania, and by including clinical samples, allowing for a comparison with asymptomatic samples.

The two studies made overall very similar findings, including strong small-scale population structure, related infections on Zanzibar and the mainland, near-clonal expansion on Pemba, and frequency of markers of drug resistance.

Strengths:

The overall results show a clear pattern of population structure. The finding of highly related infections detected in close proximity shows local transmission and can possibly be leveraged for targeted control.

Comments on revised version:

The authors have addressed my comments.

---

## [Author Response]

The following is the authors’ response to the original reviews.

**eLife assessment**
Connelly and colleagues provide convincing genetic evidence that importation from mainland Tanzania is a major source of *Plasmodium falciparum* lineages currently circulating in Zanzibar. This study also reveals ongoing local malaria transmission and occasional near-clonal outbreaks in Zanzibar. Overall, this research highlights the role of human movements in maintaining residual malaria transmission in an area targeted for intensive control interventions over the past decades and provides valuable information for epidemiologists and public health professionals.
**Reviewer #1 (Public Review):**
Zanzibar archipelago is close to achieving malaria elimination, but despite the implementation of effective control measures, there is still a low-level seasonal malaria transmission. This could be due to the frequent importation of malaria from mainland Tanzania and Kenya, reservoirs of asymptomatic infections, and competent vectors. To investigate population structure and gene flow of *P. falciparum* in Zanzibar and mainland Tanzania, they used 178 samples from mainland Tanzania and 213 from Zanzibar that were previously sequenced using molecular inversion probes (MIPs) panels targeting single nucleotide polymorphisms (SNPs). They performed Principal Component Analysis (PCA) and identity by descent (IBD) analysis to assess genetic relatedness between isolates. Parasites from coastal mainland Tanzania contribute to the genetic diversity in the parasite population in Zanzibar. Despite this, there is a pattern of isolation by distance and microstructure within the archipelago, and evidence of local sharing of highly related strains sustaining malaria transmission in Zanzibar that are important targets for interventions such as mass drug administration and vector control, in addition to measures against imported malaria.
**Strengths:**
This study presents important samples to understand population structure and gene flow between mainland Tanzania and Zanzibar, especially from the rural Bagamoyo District, where malaria transmission persists and there is a major port of entry to Zanzibar. In addition, this study includes a larger set of SNPs, providing more robustness for analyses such as PCA and IBD. Therefore, the conclusions of this paper are well supported by data.
**Weaknesses:**
Some points need to be clarified:(1) SNPs in linkage disequilibrium (LD) can introduce bias in PCA and IBD analysis. Were SNPs in LD filtered out prior to these analyses?

Thank you for this point. We did not filter SNPs in LD prior to this analysis. In the PCA analysis in Figure 1, we did restrict to a single isolate among those that were clonal (high IBD values) to prevent bias in the PCA. In general, disequilibrium is minimal only over small distances <5-10kb without selective forces at play. This is much less than the average spacing of the markers in the panel. If there is minimal LD, the conclusions drawn on relative levels and connections at high IBD are unlikely to be confounded by any effects of disequilibrium.

( 2) Many IBD algorithms do not handle polyclonal infections well, despite an increasing number of algorithms that are able to handle polyclonal infections and multiallelic SNPs. How polyclonal samples were handled for IBD analysis?

Thank you for this point. We added lines 157-161 to clarify. This section now reads:

“To investigate genetic relatedness of parasites across regions, identity by descent (IBD) estimates were assessed using the within sample major alleles (coercing samples to monoclonal by calling the dominant allele at each locus) and estimated utilizing a maximum likelihood approach using the inbreeding_mle function from the MIPanalyzer package (Verity et al., 2020). This approach has previously been validated as a conservative estimate of IBD (Verity et al., 2020).”

Please see the supplement in (Verity et al., 2020) for an extensive simulation study that validates this approach.

**Reviewer #1 (Recommendations For The Authors):**
(3) I think Supplementary Figures 8 and 9 are more visually informative than Figure 2.

Thank you for your response. We performed the analysis in Figure 2 to show how IBD varies between different regions and is higher within a region than between.

**Reviewer #2 (Public Review):**
This manuscript describes *P. falciparum* population structure in Zanzibar and mainland Tanzania. 282 samples were typed using molecular inversion probes. The manuscript is overall well-written and shows a clear population structure. It follows a similar manuscript published earlier this year, which typed a similar number of samples collected mostly in the same sites around the same time. The current manuscript extends this work by including a large number of samples from coastal Tanzania, and by including clinical samples, allowing for a comparison with asymptomatic samples.The two studies made overall very similar findings, including strong small-scale population structure, related infections on Zanzibar and the mainland, near-clonal expansion on Pemba, and frequency of markers of drug resistance. Despite these similarities, the previous study is mentioned a single time in the discussion (in contrast, the previous research from the authors of the current study is more thoroughly discussed). The authors missed an opportunity here to highlight the similar findings of the two studies.

Thank you for your insights. We appreciated the level of detail of your review and it strengthened our work. We have input additional sentences on lines 292-295, which now reads:

“A recent study investigating population structure in Zanzibar also found local population microstructure in Pemba (Holzschuh et al., 2023). Further, both studies found near-clonal parasites within the same district, Micheweni, and found population microstructure over Zanzibar.”

Strengths:The overall results show a clear pattern of population structure. The finding of highly related infections detected in close proximity shows local transmission and can possibly be leveraged for targeted control.Weaknesses:A number of points need clarification:(1) It is overall quite challenging to keep track of the number of samples analyzed. I believe the number of samples used to study population structure was 282 (line 141), thus this number should be included in the abstract rather than 391. It is unclear where the number 232 on line 205 comes from, I failed to deduct this number from supplementary table 1.

Thank you for this point. We have included 282 instead of 391 in the abstract. We added a statement in the results at lines 203-205 to clarify this point, which now reads:

“PCA analysis of 232 coastal Tanzanian and Zanzibari isolates, after pruning 51 samples with an IBD of greater than 0.9 to one representative sample, demonstrates little population differentiation (Figure 1A).”

(2) Also, Table 1 and Supplementary Table 1 should be swapped. It is more important for the reader to know the number of samples included in the analysis (as given in Supplementary Table 1) than the number collected. Possibly, the two tables could be combined in a clever way.

Thank you for this advice. Rather than switch to another table altogether, we appended two columns to the original table to better portray the information (see Table 1).

Methods(3) The authors took the somewhat unusual decision to apply K-means clustering to GPS coordinates to determine how to combine their data into a cluster. There is an obvious cluster on Pemba islands and three clusters on Unguja. Based on the map, I assume that one of these three clusters is mostly urban, while the other two are more rural. It would be helpful to have a bit more information about that in the methods. See also comments on maps in Figures 1 and 2 below.

Cluster 3 is a mix of rural/urban while the clusters 2, 4 and 5 are mostly rural. This analysis was performed to see how IBD changes in relation to local context within different regions in Zanzibar, showing that there is higher IBD within locale than between locale.

(4) Following this point, in Supplemental Figure 5 I fail to see an inflection point at K=4. If there is one, it will be so weak that it is hardly informative. I think selecting 4 clusters in Zanzibar is fine, but the justification based on this figure is unclear.

The K-means clustering experiment was used to cluster a continuous space of geographic coordinates in order to compare genetic relatedness in different regions. We selected this inflection point based on the elbow plot and based the number to obtain sufficient subsections of Zanzibar to compare genetic relatedness. This point is added to the methods at lines 174-178, which now reads:

“The K-means clustering experiment was used to cluster a continuous space of geographic coordinates in order to compare genetic relatedness in different regions. We selected K = 4 as the inflection point based on the elbow plot (Supplemental Figure 5) and based the number to obtain sufficient subsections of Zanzibar to compare genetic relatedness.”

(5) For the drug resistance loci, it is stated that "we further removed SNPs with less than 0.005 population frequency." Was the denominator for this analysis the entire population, or were Zanzibar and mainland samples assessed separately? If the latter, as for all markers <200 samples were typed per site, there could not be a meaningful way of applying this threshold. Given data were available for 200-300 samples for each marker, does this simply mean that each SNP needed to be present twice?

Population frequency is calculated based on the average within sample allele frequency of each individual in the population, which is an unbiased estimator. Within sample allele frequency can range from 0 to 1. Thus, if only one sample has an allele and it is at 0.1 within sample frequency, the population allele frequency would be 0.1/100 = 0.001. This allele is removed even though this would have resulted in a prevalence of 0.01. This filtering is prior to any final summary frequency or prevalence calculations (see MIP variant Calling and Filtering section in the methods). This protects against errors occurring only at low frequency.

Discussion:(6) I was a bit surprised to read the following statement, given Zanzibar is one of the few places that has an effective reactive case detection program in place: "Thus, directly targeting local malaria transmission, including the asymptomatic reservoir which contributes to sustained transmission (Barry et al., 2021; Sumner et al., 2021), may be an important focus for ultimately achieving malaria control in the archipelago (Björkman & Morris, 2020)." I think the current RACD program should be mentioned and referenced. A number of studies have investigated this program.

Thank you for this point. We have added additional context and clarification on lines 275-280, which now reads:

“Thus, directly targeting local malaria transmission, including the asymptomatic reservoir which contributes to sustained transmission (Barry et al., 2021; Sumner et al., 2021), may be an important focus for ultimately achieving malaria control in the archipelago (Björkman & Morris, 2020). Currently, a reactive case detection program within index case households is being implemented, but local transmission continues and further investigation into how best to control this is warranted (Mkali et al. 2023).”

(7) The discussion states that "In Zanzibar, we see this both within and between shehias, suggesting that parasite gene flow occurs over both short and long distances." I think the term 'long distances' should be better defined. Figure 4 shows that highly related infections rarely span beyond 20-30 km. In many epidemiological studies, this would still be considered short distances.

Thank you for this point. We have edited the text at lines 287-288 to indicate that highly related parasites mainly occur at the range of 20-30km, which now reads:

“In Zanzibar, highly related parasites mainly occur at the range of 20-30km.”

(8) Lines 330-331: "Polymorphisms associated with artemisinin resistance did not appear in this population." Do you refer to background mutations here? Otherwise, the sentence seems to repeat lines 324. Please clarify.

We are referring to the list of Pfk13 polymorphisms stated in the Methods from lines 146-148. We added clarifying text on lines 326-329:

“Although polymorphisms associated with artemisinin resistance did not appear in this population, continued surveillance is warranted given emergence of these mutations in East Africa and reports of rare resistance mutations on the coast consistent with spread of emerging Pfk13 mutations (Moser et al., 2021). “

(9) Line 344: The opinion paper by Bousema et al. in 2012 was followed by a field trial in Kenya (Bousema et al, 2016) that found that targeting hotspots did NOT have an impact beyond the actual hotspot. This (and other) more recent finding needs to be considered when arguing for hotspot-targeted interventions in Zanzibar.

We added a clarification on this point on lines 335-345, which now reads:

“A recent study identified “hotspot” shehias, defined as areas with comparatively higher malaria transmission than other shehias, near the port of Zanzibar town and in northern Pemba (Bisanzio et al., 2023). These regions overlapped with shehias in this study with high levels of IBD, especially in northern Pemba (Figure 4). These areas of substructure represent parasites that differentiated in relative isolation and are thus important locales to target intervention to interrupt local transmission (Bousema et al., 2012). While a field cluster-randomized control trial in Kenya targeting these hotspots did not confer much reduction of malaria outside of the hotspot (Bousema et al. 2016), if areas are isolated pockets, which genetic differentiation can help determine, targeted interventions in these areas are likely needed, potentially through both mass drug administration and vector control (Morris et al., 2018; Okell et al., 2011). Such strategies and measures preventing imported malaria could accelerate progress towards zero malaria in Zanzibar.”

Figures and Tables:(10) Table 2: Why not enter '0' if a mutation was not detected? 'ND' is somewhat confusing, as the prevalence is indeed 0%.

Thank you for this point. We have put zero and also given CI to provide better detail.

(11) Figure 1: Panel A is very hard to read. I don't think there is a meaningful way to display a 3D-panel in 2D. Two panels showing PC1 vs. PC2 and PC1 vs. PC3 would be better. I also believe the legend 'PC2' is placed in the wrong position (along the Y-axis of panel 2).Supplementary Figure 2B suffers from the same issue.

Thank you for your comment. A revised Figure 1 and Supplemental Figure 2 are included, where there are separate plots for PC1 vs. PC2 and PC1 vs. PC3.

(12) The maps for Figures 1 and 2 don't correspond. Assuming Kati represents cluster 4 in Figure 2, the name is put in the wrong position. If the grouping of shehias is different between the Figures, please add an explanation of why this is.

Thank you for this point. The districts with at least 5 samples present are plotted in the map in Figure 1B. In Figure 2, a totally separate analysis was performed, where all shehias were clustered into separate groups with k-means and the IBD values were compared between these clusters. These maps are not supposed to match, as they are separate analyses. Figure 1B is at the district level and Figure 2 is clustering shehias throughout Zanzibar.

The figure legend of Figure 1B on lines 410-414 now reads:

“(B) A Discriminant Analysis of Principal Components (DAPC) was performed utilizing isolates with unique pseudohaplotypes, pruning highly related isolates to a single representative infection. Districts were included with at least 5 isolates remaining to have sufficient samples for the DAPC. For plotting the inset map, the district coordinates (e.g. Mainland, Kati, etc.) are calculated from the averages of the shehia centroids within each district.”

The figure legend of Figure 2 on lines 417-425 now reads:

“Figure 2. Coastal Tanzania and Zanzibari parasites have more highly related pairs within their given region than between regions. K-means clustering of shehia coordinates was performed using geographic coordinates all shehias present from the sample population to generate 5 clusters (colored boxes). All shehias were included to assay pairwise IBD between differences throughout Zanzibar. Pairwise comparisons of within cluster IBD (column 1 of IBD distribution plots) and between cluster IBD (column 2-5 of IBD distribution plots) was done for all clusters. In general, within cluster IBD had more pairwise comparisons containing high IBD identity.”

(13) Figure 2: In the main panel, please clarify what the lines indicate (median and quartiles?). It is very difficult to see anything except the outliers. I wonder whether another way of displaying these data would be clearer. Maybe a table with medians and confidence intervals would be better (or that data could be added to the plots). The current plots might be misleading as they are dominated by outliers.

Thank you for this point and it greatly improved this figure. We changed the plotting mechanisms through using a beeswarm plot, which plots all pairwise IBD values within each comparison group.

(14) In the insert, the cluster number should not only be given as a color code but also added to the map. The current version will be impossible to read for people with color vision impairment, and it is confusing for any reader as the numbers don't appear to follow any logic (e.g. north to south).

Thank you very much for these considerations. We changed the color coding to a color blind friendly palette and renamed the clusters to more informative names; Pemba, Unguja North (Unguja_N), Unguja Central (Unguja_C), Unguja South (Unguja_S) and mainland Tanzania (Mainland).

(15) The legend for Figure 3 is difficult to follow. I do not understand what the difference in binning was in panels A and B compared to C.

Thank you for this point. We have edited the legend to reflect these changes. The legend for Figure 3 on lines 427-433 now reads:

“Figure 3. Isolation by distance is shown between all Zanzibari parasites (A), only Unguja parasites (B) and only Pemba parasites (C). Samples were analyzed based on geographic location, Zanzibar (N=136) (A), Unguja (N=105) (B) or Pemba (N=31) (C) and greater circle (GC) distances between pairs of parasite isolates were calculated based on shehia centroid coordinates. These distances were binned at 4km increments out to 12 km. IBD beyond 12km is shown in Supplemental Figure 8. The maximum GC distance for all of Zanzibar was 135km, 58km on Unguja and 12km on Pemba. The mean IBD and 95% CI is plotted for each bin.”

(16) Font sizes for panel C differ, and it is not aligned with the other panels.

Thank you for pointing this out. Figure 3 and Supplemental Figure 10 are adjusted with matching formatting for each plot.

(17) Why is Kusini included in Supplemental Figure 4, but not in Figure 1?

In Supplemental Figure 4, all isolates were used in this analysis and isolates with unique pseudohaplotypes were not pruned to a single representative infection. That is why there are additional isolates in Kusini. The legend for Supplemental Figure 4 now reads:

“Supplemental Figure 4. PCA with highly related samples shows population stratification radiating from coastal Mainland to Zanzibar. PCA of 282 total samples was performed using whole sample allele frequency (A) and DAPC was performed after retaining samples with unique pseudohaplotypes in districts that had 5 or more samples present (B). As opposed to Figure 1, all isolates were used in this analysis and isolates with unique pseudohaplotypes were not pruned to a single representative infection.”

(18) Supplemental Figures 6 and 7: What does the width of the line indicate?

The sentence below was added to the figure legends of Supplemental Figures 6 and 7 and the legends of each network plot were increased in size:

“The width of each line represents higher magnitudes of IBD between pairs.”

(19) What was the motivation not to put these lines on the map, as in Figure 4A? This might make it easier to interpret the data.

Thank you for this comment. For Supplemental Figure 8 and 9, we did not put these lines that represent lower pairwise IBD to draw the reader's attention to the highly related pairs between and within shehias.

**Reviewer #2 (Recommendations For The Authors):**
(1) There is a rather long paragraph (lines 300-323) on COI of asymptomatic infections and their genetic structure. Given that the current study did not investigate most of the hypotheses raised there (e.g. immunity, expression of variant genes), and the overall limited number of asymptomatic samples typed, this part of the discussion feels long and often speculative.

Thank you for your perspective. The key sections highlighted in this comment, regarding immunity and expression of variant genes, were shortened. This section on lines 300-303 now reads:

“Asymptomatic parasitemia has been shown to be common in falciparum malaria around the globe and has been shown to have increasing importance in Zanzibar (Lindblade et al., 2013; Morris et al., 2015). What underlies the biology and prevalence of asymptomatic parasitemia in very low transmission settings where anti-parasite immunity is not expected to be prevalent remains unclear (Björkman & Morris, 2020).”

(2) As a detail, line 304 mentions "few previous studies" but only one is cited. Are there studies that investigated this and found opposite results?

Thank you for this comment. We added additional studies that did not find an association between clinical disease and COI. These changes are on lines 303-308, which now reads:

“Similar to a few previous studies, we found that asymptomatic infections had a higher COI than symptomatic infections across both the coastal mainland and Zanzibar parasite populations (Collins et al., 2022; Kimenyi et al., 2022; Sarah-Matio et al., 2022). Other studies have found lower COI in severe vs. mild malaria cases (Robert et al., 1996) or no significant difference between COI based on clinical status (Earland et al. 2019; Lagnika et al. 2022; Conway et al. 1991; Kun et al. 1998; Tanabe et al. 2015)”

(3) Table 2: Percentages need to be checked. To take one of several examples, for Pfk13-K189N a frequency of 0.019 for the mutant allele is given among 137 samples. 2/137 equals to 0.015, and 3/137 to 0.022. 0.019 cannot be achieved. The same is true for several other markers. Possibly, it can be explained by the presence of polyclonal infections. If so, it should be clarified what the total of clones sequenced was, and whether the prevalence is calculated with the number of samples or number of clones as the denominator.

Thank you for this point. We mistakenly reported allele frequency instead of prevalence. An updated Table 2 is now in the manuscript. The method for calculating the prevalence is now at lines 148-151:

“Prevalence was calculated separately in Zanzibar or mainland Tanzania for each polymorphism by the number of samples with alternative genotype calls for this polymorphism over the total number of samples genotyped and an exact 95% confidence interval was calculated using the Pearson-Klopper method for each prevalence.”